# Identification of Risk Factors in Business Valuation

**Muhammad Najib Razali [1,*], Rohaya Abdul Jalil [2], Kamalahasan Achu [2] and Hishamuddin Mohd Ali [2]**

[1]  Faculty of Built Environment and Surveying, Centre of Environmental Sustainability and Water Security, Universiti Teknologi Malaysia, Johor Bahru 81310, Malaysia
[2]  Faculty of Built Environment and Surveying, Universiti Teknologi Malaysia, Johor Bahru 81310, Malaysia; rohaya@utm.my (R.A.J.); kamalahasan@utm.my (K.A.); hishamuddin@utm.my (H.M.A.)
[*]  Correspondence: mnajibmr@utm.my

**Abstract:** It is widely accepted that risk and uncertainty are integral parts of the property valuation process. Uncertainty in property valuation is derived from the characteristics of property itself. The issue pertaining to risk and uncertainty in property valuations is currently one of the key concerns in global valuation practice to date in addressing the decision of risk and uncertainty in valuation, especially for business purposes or in the current term known as business valuation. The judgment and experience still depend on the expertise of the individual valuers alone. The valuation methods used can cause problems if certain elements in business such as risk are highlighted, especially to determine market value. There is a need for valuers to express assumptions which take into account risk and uncertainties, and then pass on the results of the estimation process to the end user of the valuation report. This research employed Analytical Hierarchical Process (AHP) to identify the level of risk in business valuation for valuers to identify which risk areas will expose them to professional liabilities, which then leads to mitigation of risk to determine value in business valuations. AHP will also be able to identify the level of risk in each of the approaches in business valuation which could help valuers to determine the value and market value in the valuation process. This paper will propose some practical approaches of how to address the risk and uncertainty of the valuation process, especially for the purpose of business valuation.

**Keywords:** valuation; risk; business; AHP

## 1. Introduction

Business valuation has been conventionally based on the financial and monetary representation of a business element by using financial, economic, accounting evidence, and materials. The process is not directly assessed by the performance of the business practices and companies' operational aspects. The use of additional information with regard to the business valuation process requires the development of a new process which is able to handle different types of information. Business valuation also needs more data which could be developed from historical and current performance, forecasting, and strategic focuses. This information will be used as a strong point to develop a financial impact where it can be incorporated to determine value in the business valuation process. The main objective in business valuation is to determine a representation of the overall worth of a business entity. The value of a business is from a monetary point of view. There are several factors that business valuation requires such as: merger and acquisition, acquiring a business which needs to determine the price to pay, brand and equity valuation, dividing assets between individuals, and the assets' value and asset class worth. A precise representation of the overall value of the company is the key to ensuring a good, strategic decision in the business valuation process.

Business valuation refers to the establishment of value of an enterprise which is planning to continue the business and demands the assessment of the business which employs forecasted data in the calculations. Businesses have evolved recently and need to face

several challenges and unprecedented circumstances. The business dynamic environment where risk turns to volatility means that business values may fluctuate in a rather wide range. Risk and uncertainty need to be handled systematically due to changes of the drivers. According to Kunas (2012), the fluctuation of business value within a wide range predetermines the problematic aspects of rendering the final conclusions on business value. Risk valuation is very important and is a sophisticated element within business valuation. This is due to the complication of reflecting the risk factor in valuation in business and investment projects. According to Corrado et al. (2001), the value of a company varies directly when the level of profit and expected growth of cash flow changes, and changes are conversely compared with changes in risk. Within the cash flow, to assess risk requires the expected value to be discounted in considering the required rate of return, which depends on the riskiness of the property. This risk could relate to a negative value of the business valuation which means it would not generate the predicted cash flow. Consequently, it will decrease the business value. Although risk that is associated with business valuation in order to determine market value is still widely discussed in the scientific literature, there is no mitigation in determining risk classifications for the purpose of business valuation.

In the valuation process, it has been accepted that there will be risk and uncertainty. According to Sloman (1995), uncertainty is when the outcome may or may not occur, and the probability of occurring is ambiguous. Epstein (1999) stated that risk is where the perceived likelihood of events of interest is represented by probabilities, whereas uncertainty signifies circumstances where information available needs to be measured through probabilities. Therefore, in the valuation process, valuers need to identify all the possibilities that relate to the process in determining the value. Furthermore, as the subject of the valuation is property, as such, the characteristics and process of the property will trigger the possibilities of risk. This includes land and cost, finance, construction, the development process, supply and demand, macroeconomic and microeconomic factors, and socioeconomics. These factors need to be assessed individually during the valuation process in order to determine the risk that could be associated with the value.

Mitigating risk liability in the business valuation process will present a logical course for the valuation model by identification of business value drivers. A conceptual scheme to establish systematic business value will depend on the classification of business value drivers. The methodology will be based on decomposition of business value, methods, and drivers. The classification of drivers will be able to form the principle of business value decomposition which enables more detailed value driver classifications. Risk management classifications from business value drivers will create characteristics of risk in business valuations. A variety of factors detail the very complex classification of risks. According to Kazlauskienė and Christauskas (2007), the classification is based on the nature of factors determining risk where the external environment causes external or systematic risk, and factors of the internal environment cause unsystematic risk. Furthermore, systematic risk does not depend on a company's activities or financial operations. In order to highlight issues in business valuation with regards to risk, uncertainties, and the determination of the fair market value, it is important that the solution has a specific model that elucidates these conditions which have different business structures and are better at mitigating the liabilities.

Risk management in business can be regarded as an ongoing and iterative process. It has its own characteristics which need to be identified by valuers and professionals. Therefore, this research aims to highlight these issues by using research tools to quantify all the risk involved in determining the business valuation. In addition, this study also examined how professional liability can mitigate upon deriving fair market value in business valuation.

## 2. Literature Review

Business valuation has many complexities and controversial facets for a business valuation expert, especially a valuer. The valuer must have passion for the subject and be

willing to accept all odds in pursuing the objective. The valuer also needs to have in-depth knowledge of accounting, corporate governance implications, financial analysis, industry information, management knowledge, and standards for valuing a business. Property is always linked to the business entity; therefore, the role of valuers in business has been significant in recent years. Business valuation is frequently a confusing subject. Some do not know the purpose of a valuation, such as not realising what the valuations are required for. The valuers need to tackle this problem hands-on if the case is really pursuing the subject matter, which could also mitigate risk in professional liabilities of a business valuation. Moreover, the selection of an appropriate technique is of paramount importance in arriving at the value of a business. The three universal approaches for business valuation are: asset approach, income approach, and market approach; therefore, the question is which approach a valuer shall consider for valuing a business. Valuers are faced with the task of valuing business equity and have to choose an approach, especially if the intrinsic value does not appear in the financial statements. Some elements in business valuation such as intangible value, multiples, or discounted cash flows are easy to use, but they could also be misused. Therefore, relevant facts must be used in order to achieve an accurate value.

Generally in the valuation process, it has long been debated that risk and uncertainty are due to the interchangeability often found in another description. Lorenz and Lützkendorf (2011) define risk from the perspective of investment and finance associated with an asset that contains volatility of its returns. French and Gabrielli (2004) describe risk as the measurement of a loss identified as a possible outcome of the decision and uncertainty as anything that is not known about the outcome of a venture at the time when the decision is made. While Adair and Hutchison (2005) define risk from the context of property investment as the probability that a target rate of return will not be realised and argue that the concept of risk supposes that all outcomes together with their probabilities of occurrence are known. Adams et al. (1998) argues that risk is a word that refers to the future where no objective existence with the future is only by imagination. From the definitions, it seems risk is associated with something uncertain. Nevertheless, in business valuation, where money and profit are important, error in risk is crucial where industry players need to minimise the element of uncertainty to determine risk in the business valuation process. Industry players prefer the element of risk and uncertainty to be dealt with. French and Gabrielli (2004) propose that the most appropriate measure to manage risk and uncertainty in the valuation process is through a statistical approach. RICS (1994) also emphasise that valuers should draw attention to and comment on any issues affecting the uncertainty of the valuation. Furthermore, valuers are responsible to provide explicit risk communication to clients in the form of a quantifiable risk score which depends on the valuation case subject matter. Communication of risk in valuations assumes far more importance in emerging markets where the data on comparable evidence for basing the valuation are sparsely available, and assumptions need to be clearly reported (Gupta and Tiwari 2014). In addition, risk reporting alongside property valuations assumes greater importance in the background, especially with recent financial crises such as the Global Financial Crisis (GFC). Banking sectors are also getting strict with valuations that were determined by the valuer. The valuations of the cases provided by the valuers are normally based on current market conditions. Nevertheless, the decisions that are made are always exposed to risk from the element of uncertainty. As mentioned in RICS (1994), valuation reports must not be misleading or create a false impression. The valuer should draw attention to and comment on any issues affecting uncertainty of the valuation. The extent of that commentary will vary depending on the purpose of the valuation and the format of the report agreed upon with the client.

In general, any profession or employers may be held liable for an accident arising out of the general course of employment. As a professional, it is necessary to take every step possible to try to prevent lawsuits by following professional standards and duty of care; however, we live in a litigious society where just about anyone can sue for negligence. Risk in professional practices such as values also involves professional liability. Professional

liability is a legal obligation arising out of a professional's error, negligent acts, or omissions during the course of a valuation. In the process to determine the value of a property, in most cases, it is carried out very well; nevertheless, in some cases, valuers might provide value incorrectly. Consequently, in order to achieve the most accurate value in the valuation process, valuers have a duty of care to take all measures when surveying or valuing a property. This includes the inspection process which requires full and detailed surveys. As a result, the final valuation result will be a professional opinion of value which could be different from one professional to another. The differences among valuers in determining value, known as margin error, depend on the type of property. This is where evidence will play a major role to prove that valuers did not act in a negligent manner.

The future is unknown; thus, that is where risk and uncertainty exist. According to Aven (2016) risk has existed for 2400 years by the Athenians; however, risk assessment and risk management were created around 30 to 40 years ago. There are many definitions of risk. Based on previous studies, risk is defined as undesirable, negative effects, unfortunate occurrences, deviation, or unexpected occurrences from the planning of the activity. There are also many other risk definitions. Kliem and Ludin (2019) categorised risk as follows:

i.     Acceptable vs. non-acceptable risks
ii.    Short-term vs. long-term risks
iii.   Positive vs. negative risks
iv.    Manageable vs. non-manageable risks
v.     Internal vs. external risks

Usilappan (2006), Williams et al. (1998) defined risk from an investment value perspective, which is different from the targeted returns from investment outcomes, such as when risk has no certainty of the outcome. Furthermore, risk is an objective concept and can be measured. Khumpaisal (2011), Wiegelmann (2012) defined risk as a possibility of negative or unfavourable impacts from a present process or future event to an asset, project, or some element of value. Riley et al. (2006) explained that risk is also known as uncertainty. According to Manaf et al. (2006), risk is part of business and public life and is an adverse event based on circumstances. From an investment view, investment objectives cannot be achieved as risk returns uncertainty over time. Risk also arises from unexpected volatility in asset returns over time or the uncertainty of future outcomes. According to the Institute of Risk Management (2002), risk is a combination of the probability of an event. Risk is also a consequence that constitutes the upside (benefit) or downside (threat to success) on one or more project objectives, i.e., scope, schedule, cost, and quality. Sloman and Wride (2009), Obinna (2017) defined risk a result of an action that may or may not occur. Risk as defined by Kliem and Ludin (2019) is the occurrence of an event that has been impacted on, and there are five elements of risk, as follows:

i.     The possibility of the occurrence of risks whether high, medium, or low risks.
ii.    Frequency of occurrence of risks.
iii.   Impact of occurrence of risk.
iv.    Other important related factors associated with risks.
v.     The exposure of the impact of risk on the product or result of the project.

Riley et al. (2006) further elaborated on sources of risk which may cause fluctuations, as follows:

i.     Fluctuations in expected income—varying dividends, missed interest payments, or unoccupied rental of real estate.
ii.    Fluctuations in the expected future price of the asset—changing economic conditions or asset-specific circumstances.
iii.   Fluctuations in the amount available for reinvestment and fluctuations in returns earned from reinvestment—changes in tax rates, interest rates, or asset returns.

Wilkinson and Reed (2008), Riley et al. (2006), Williams et al. (1998) classified risk into systematic risk (involves movement in a group over time) and unsystematic risk

(deviation of individual security returns from group average). Details of both systematic and unsystematic risk are as follows:

i.      Systematic risk

The risk of holding a diversified portfolio is minimum, and systematic risk cannot be reduced by having diversification. The remaining risk is called systematic risk. This risk affects the economic or financial system (pervasive throughout the economy) such as interest rates, economic growth rate, changes in taxes, changes in government policies, fluctuation of exchange rates, and major military actions.

ii.     Unsystematic risk

In contrast with systematic risk, unsystematic risk can be reduced by diversification. This risk is also known as asset-specific risk. Unsystematic risk is focusing on a smaller view rather than a broad economic view such as the type of asset or security issuer. The examples of unsystematic risk are poor management decisions, labour strikes, deterioration of product or service quality, and the rise of new competitors.

Muka (2017) explained that risk is an interaction between threat and hazard of vulnerabilities. When risk is present, outcomes cannot be forecasted with certainty; thus, risk gives rise to uncertainty. As the future is unknown, risk and uncertainty exist. According to Tesfaye et al. (2016) risk and uncertainty are two common terms which are closely related with a negative outcome of a certain event. There are many ways of defining the relationship between risk and uncertainty. Risk and uncertainty are two common terms in the risk management literature. Kliem and Ludin (2019) explained that risk will reflect uncertainty, while Williams et al. (1998) defined that uncertainty arises when we perceive risk and doubt to predict the future. Sloman and Wride (2009) explained that uncertainty exists when the possibility of the outcome is unknown. The contrast of uncertainty is certainty. Different to uncertainty, certainty is a subjective concept; thus, it cannot be measured directly; however, certainty can be measured indirectly, such as in science and mathematics.

*Risk in Business Valuation*

In order to determine the risk levels in business valuations, the main research instrument to investigate risk is through a questionnaire. For the purpose of this objective, the questionnaire is categorised under two different sections, namely, risk in a business valuer's profession and risk in the business valuation process. Under the section of risk in a business valuer's profession, the aim is to identify the areas where valuers are able to limit the exposure of professional liabilities among business valuation practitioners. Under this category, respondents are also required to categorise risk, whether it is systematic or unsystematic risk. The explanations of these risk categories were explained in a previous chapter. These questions need the valuer's professional views based on their experience as valuers, accountants, or institutional investors. The risk areas in business valuation are based on three major risk categories, namely, risk in business valuation, systematic and unsystematic risk, and risk in quality valuation reporting. These risk areas were identified from previous research undertaken by Trugman (2016). The risk areas were categorised based on the approach or method used in business valuations. In an asset-based approach, the most likely risk to be encountered is when adjustments to assets typically require adjustments to the income statement in the form of increased expenses or reduced revenues. In the process of determining the value, managers have incentives to deflate reported earnings which occur when a firm is performing very well and encourage them to put some funds away for a rainy day. Furthermore, accounting rules could also lead to the understatement of assets. In several countries, accounting standards require firms to record expense outlays for research and investigation should they encounter future valuations for owners which could lead to uncertain outcomes.

Asset understatement could occur when managers have incentives to understate liabilities. Accountants need to analyse whether managers have a tendency to understate or overstate assets and, if necessary, adjust the balance sheet and income statement accordingly.

This will also lead to the ambiguity in determining a fair value. In an income approach, the evaluation of a firm for the purpose of business valuation will involve larger investment processes that include several elements, namely, establishing the objectives of the valuation, forming expectations about the future returns, and combining income into portfolios to maximise progress toward the investment objectives. Furthermore, the foundation is projecting returns and assessing risk to identify mispricing of the case subject matter in the hope of generating returns that more than compensate the investor for risk. For valuers who do not have a comparative advantage in identifying mispricing of the case subject matter, the focus should be on gaining an appreciation for how the income approach would affect the risk of a given portfolio and whether it fits the profile that the portfolio is designed to maintain. Unlike assets which have fundamental existence, income analysis in business valuation is relative. In analysing income, the monetary amount representing the difference between revenues and expenses (or, in other words, revenues minus expenses) equals earnings or income. Therefore, in this approach, a financial statement is vital and is used to investigate the revenues and expenses to understand income. It could also benefit from understanding important items in the income approach, namely, revenues, expenses, and earnings related to underlying assets and liabilities. Another important element in the income approach is income statement adjustments. The normalising adjustment and control adjustment will turn the income statement into a proper application. In normalising adjustment, the earnings of the company need to be reasonably well-run based on an equivalent basis. It can be further divided into two types, namely, typical financial buyers and particular buyers. The majority of valuers do not distinguish between normalising and control adjustment. It is important to distinguish between types of income statement adjustments when determining the discount rate applicable to derived earnings. The discount rate or capitalisation rate applied to a particular measure or earnings must be appropriate to measure net income, pre-tax income, debt-free income, or another level of the income statement (Mercer and Harms 2020). These are the possible areas which could influence the justification of value in an income approach.

In a market approach, the analysis involves public company information which aims to obtain and analyse the financial and operating data of the companies. Values will use the available information in order to ensure that the property subject will compare accordingly. For public companies, it is important to make some adjustment to ensure the data were adjusted based on the valuation purpose. Nevertheless, some of the information needed is not available which indicates that limitation of data is one of the risks that needs to be taken into account. This includes comparability adjustment and the nonrecurring adjustment between listed and closely held companies in several items. As a result, valuers need to proceed with normalisation which places valuers in the risk areas.

## 3. Methods and Materials

AHP is the analysis technique developed by Saaty (2008) and uses the concept of multi-criteria decision-making (MCDM) methods, which has been successful in various practical decision-making problems. This method is widely used in many fields including engineering, real estate, economics, administration, and management at different levels (Ferreira et al. 2019; Rao 2021; Salo et al. 2021; Siejka 2015; Wu et al. 2012; Yoxas et al. 2011; Zelenović Vasiljević et al. 2012). This method is one of the important parts of modern decision analysis theory that evaluates a series of alternatives which aim to find the optimal plan in decision-making (Sun et al. 2021). Furthermore, AHP has been considered as a mathematical tool especially for policymakers to make wise choices based on several diverse factors (Doumpos and Zopounidis 2004). According to Malczewski (2006), AHP has three important procedures, namely, to use known alternatives, and then multiple rules are used as the basis for evaluation and decision-makers to express their preference information, and arrange the sequence of alternatives. AHP will determine the priorities for all of the identified variables based on the decision criterion and priorities for each criterion according to the objectives. The gathering of priorities with their unique needs

will be the most responsible option, and the proportion of the gathered priorities will show the relative qualities in achieving the objective (Deep et al. 2018).

This research aims to identify the level of risk in business valuation in order for valuers to be able to identify which risk areas will expose them to professional liabilities. Consequently, this will lead to mitigation of risk in determining value in business valuation. AHP will also be able to identify the level of risk in each of the approaches in business valuation which could help valuers to determine the fair value and fair market value in the valuation process. In AHP, the decision should be determined by the problem which, in this research, is to identify risk levels and professional liabilities that lead to the determination of fair value. The use of MCDM methods has been successfully applied in many practical decision-making problems. In the AHP process, complicated decision problems can be decomposed into several hierarchies which accord to the attributes or criteria.

By using AHP, the first decision problem is to determine risk in business valuations under two categories (professional liabilities and the approached valuation); then, it needs to decompose into a hierarchical structure.

Nevertheless, AHP has some issues with the inconsistency of the restriction of pairwise comparisons to a 1 to 9 scale. According to Huang et al. (2020), the problem of inconsistency is due to the problematic correspondence between the verbal and the numeric scales. In addition, there is variation in the verbal expression from one person to another as well the dependence on the type of elements involved in the comparison (Belton and Goodwin 1996).

In AHP, the 9-point integer scale as shown in Table 1 will be used to quantify the intangible attributes or criteria into measurable numerical numbers. Then, all pairwise comparisons will be arranged into comparison matrices.

**Table 1.** The 9-point Scale.

| Intensity of Importance | Definition | Explanation |
| :---: | :---: | :--- |
| 1–2 | Equally high risk | Two activities contribute equally to the objective |
| 3–4 | Low risk of one over another | Experience and judgement slightly favouring one area over another |
| 5–6 | High risk of one over another | Experience and judgement strongly favouring one area over another |
| 7–8 | Demonstrated risk | An activity is strongly favoured, and its dominance demonstrated in practice |
| 9 | Very high risk | When compromise is needed |

(Amended from Saaty 2008).

In the first step, a decision problem should be defined; then, structure the decision hierarchically by breaking down the decision problem into a hierarchy of interrelated decision elements, which usually includes three hierarchy levels: objective level, criteria level, and alternatives level. In the AHP, it is assumed that the relation of higher-level elements from lower-level elements is independent, and the elements within a level are also assumed to be independent. In the second step, elements within the same level are pairwise compared with a given criterion, which is located at the higher level. The intangible attributes are measured by a scale of absolute judgments that represent how much one element dominates another more with respect to a given attribute.

Questionnaires have been disseminated among four types of respondents: valuers, accountants, academics, and institutional investors. As the topic of this research and the areas of discussions are relatively new in the industry, therefore, the sampling technique that is chosen is the stratified sampling technique. Initially, due to the background of case studies, the selection of respondents needs to be very carefully processed. The respondents need at least to have a basic knowledge of valuations and more than 10 years' experience. This is due to the complexity of the business valuation process itself. This is especially

important for valuers and accountants who are directly involved in determining the business valuation process. Furthermore, experienced valuers are more familiar with the case study, especially in the second section of the questionnaire. As for institutional investors, the view from this professional group is essential due to the fact that this group is mostly clients of the business valuation reports. Therefore, the perception from this group will contribute significantly in assessing the ranking of the business valuation risk in terms of professional liabilities and risk mitigation. As for academics, the aim of choosing them as respondents is to obtain a theoretical background regarding the subject matter as well as for validation purposes.

The experts from the two series of FGDs have also been asked to evaluate based on the weight of factors with all areas of risk, namely, profession, method, and process. Table 2 presents the evaluation of factor weights by experts with the range between 1 (very low) to 10 (very high). The fuzzy set of mathematical expressions, which represent the degree to which that expert's opinion is set to a greater or lesser degree, is indicated by weights. These weights are represented by real-number values in the close interval between 1 and 10. The grade of weightage is expressed by the factor's function. Based on the hierarchy structure of risk factors, a panel of experts evaluates the fuzzy weights with the Likert scale choice of answers. From the results, the average fuzzy weights are calculated based on the following equation, which is based on the model developed by Sun et al. (2008):

$$
\text{Å} = \left\{ \begin{matrix} 1 & a_{\widetilde{12}} & a_{\widetilde{1n}} \\ a_{\widetilde{12}} & 1 & a_{\widetilde{2n}} \\ a_{\widetilde{n1}} & a_{\widetilde{n12}} & 1 \end{matrix} \right\} = \left\{ \begin{matrix} 1 & a_{\widetilde{12}} & a_{\widetilde{1n}} \\ 1/a_{\widetilde{12}} & 1 & a_{\widetilde{2n}} \\ 1/a_{\widetilde{n1}} & 1/a_{\widetilde{n12}} & 1 \end{matrix} \right\}
$$

where

$$
\{9^{-1}, 8^{-1}, 7^{-1}, 6^{-1}, 5^{-1}, 4^{-1}, 3^{-1}, 2^{-1}, 1^{-1}, \overline{1}, \overline{2}, \overline{3}, \overline{4}, \overline{5}, \overline{6}, \overline{7}, \overline{8}, \overline{9}\}\ 1, i \neq j,\ i = j
$$

where $i = 1, 2, \ldots . n$; $j = 1, 2$. $A_{xy}$ represents the fuzzy weight assigned to factor $c_i$, given the expert k; $A_{xy} = (a_{x,,,y}, c_{x,y}, d_{x,y})$ represents the average fuzzy of factor $c_{i,j}$.

The average fuzzy weights are calculated as follows:

$A_1 = (0.76, 0.87, 0.75)$
$A_2 = (0.52, 0.68, 0.51)$
$A_3 = (0.51, 0.72, 0.54)$
$A_4 = (0.63, 0.74, 0.78)$
$A_5 = (0.72, 0.61, 0.91)$
$A_6 = (0.72, 0.93. 0.75)$
$A_7 = (0.62, 0.64, 0.65)$
$A_8 = (0.78, 0.83, 0.61)$
$A_9 = (0.62, 0.81, 0.93)$
$A_{10} = (0.72, 0.62. 0.94)$
$A_{11} = (0.83, 0.94, 0.77)$
$A_{12} = (0.81, 0.72, 0.95)$
$A_{13} = (0.72, 0.94, 0.62)$
$A_{12} = (0.85, 0.92, 0.77)$

The score for the alternatives is on three factors: C1 (profession), C2 (method), and C3 (process). The evaluation for these factors on risk sources can be expressed from the following matrix:

$$
R_1 = \begin{matrix} 0.94 & 0.93 & 0.94 \\ 0.92 & 0.87 & 0.83 \\ 0.78 & 0.74 & 0.93 \end{matrix}
$$

The evaluation matrix can be acquired as follows:

$$B_1\,w_1\,R_1 \quad = \quad \begin{bmatrix} 0.51 & 0.54 & 0.62 \end{bmatrix} \begin{bmatrix} 0.94 & 0.93 & 0.94 \\ 0.92 & 0.87 & 0.83 \\ 0.78 & 0.74 & 0.93 \end{bmatrix}$$

$$= \quad \begin{bmatrix} 0.72 & 0.61 & 0.87 \end{bmatrix}$$

**Table 2.** The Evaluation of Factor Weights by Experts.

| Experts | Factors | | |
| --- | --- | --- | --- |
| | Profession | Method | Process |
| Expert 1 | 6 | 7 | 6 |
| Expert 2 | 5 | 6 | 5 |
| Expert 3 | 5 | 7 | 6 |
| Expert 4 | 5 | 6 | 7 |
| Expert 5 | 6 | 5 | 8 |
| Expert 6 | 6 | 8 | 6 |
| Expert 7 | 5 | 5 | 5 |
| Expert 8 | 6 | 7 | 5 |
| Expert 9 | 5 | 7 | 8 |
| Expert 10 | 6 | 5 | 8 |
| Expert 11 | 7 | 8 | 6 |
| Expert 12 | 7 | 6 | 8 |
| Expert 13 | 6 | 8 | 5 |
| Expert 14 | 7 | 8 | 6 |

*Data Collections*

In using the mixed method research, Johnson and Turner (2003) stated that a fundamental principle that must be adhered to on the part of researchers should be to collect multiple data using different strategies, approaches, and methods in such a way that the resulting mixture or combination is likely to result in complementary strengths and non-overlapping weaknesses. Therefore, an effective use of this principle is a major justification for the use of mixed methods research because the product of the combination of different methods will be superior to the use of a mono method (Johnson and Onwuegbuzie 2004). Different types of data are collected by researchers for different purposes in a research project. The research questions generally determine the type of data needed to fulfil the aims and objectives of the research. Tashakkori and Teddlie (2010) emphasised that methodological principles generally underpin the conduct of research. It is the research questions that determine the specific methods to be used and the sort of data collection strategy suitable for a particular research objective. Generally, the type of data collected will determine the sort of analysis and the interpretation of the findings that will emerge from the data.

Generally, questions used for interviews are crafted from themes or topics of interest in case studies that a researcher is studying. The questions are formulated to address research questions and objectives of a study. Interviewing questions are grouped into two: open-ended and closed-ended questions. Open-ended questions generally allow respondents the opportunity to give a broad range of answers on issues in response to questions, whereas closed-ended questions offer very limited opportunity for participants to give alternate answers to questions (Runeson and Höst 2009). However, most case study researchers use open-ended questions for obtaining information. Further, interviewees have been asked to propose insights into case studies sometimes to build on it for further enquiry. Interviews were adopted to complement the other sources of data for this research. It was an opportunity for practitioners to contribute to the research and ensure that the findings of the research were practical.

The researcher has the opportunity to ask further probing questions that emanate from the answers provided by respondents. Runeson and Höst (2009) suggested that development of the conversation dictates the order of the questions that are asked. This

method allows for extensive and in-depth exploration of the case studies and on flexibility in general and how values attached to flexibility are likely to be determined from practitioners' perspectives. In view of the use of face-to-face semi-structured interviews, the researcher had the opportunity to ask further probing questions and obtained rich in-depth knowledge on the cases to draw conclusions for the research after evaluation.

The probability sampling technique adopted for the quantitative data has the potential of ensuring that there is large enough data to enable analytical generalisations of the data to the population. Besides, every single member of the population has an equal chance of being selected to represent the population, hence, a better choice for the large quantitative data. DCF and business valuation methods are characterised by mathematical formulae for deriving results to confirm or refute the potential of business valuation models for the evaluation of flexibility in practice. The selection of participants was based on certain criteria including level of experience, qualification, position in the organisation, and responsibilities. Selected participants were well experienced and conversant with the use of property valuation models and decision making in property development and investment. The purposive sampling technique ensured that all participants had knowledge of the subject area leading to very rich and an in-depth response to the interview questions.

The participants were selected through a meticulous analysis of the most active property companies in Malaysia. Purposively, the participants selected were in senior roles such as development directors (where final decision making occurs) or in ownership of their organisations as private practitioners. Others were also senior partners in their respective organisations, thereby ensuring that those with a certain level of pedigree and knowledge participated in the research. Therefore, views expressed by participants were based on sound knowledge, experience, and good judgement, ensuring a high level of reliability in the data, resulting in valid findings and conclusions.

## 4. Results and Findings

### 4.1. Demographic Analysis

All respondents have an undergraduate education, and their working experience ranges from 10 to 30 years. As illustrated in Figure 1, most of the respondents who participated in answering the questionnaire were from the valuer's group which constituted 51% of the total respondents. This was followed by accountants and institutional investors (19% and 17%, respectively). While academics answered the questionnaire form, they only contributed 13% of the total respondents. The reason for the low percentage is that most academics in Malaysia are still not familiar with the business valuation concept and technique.

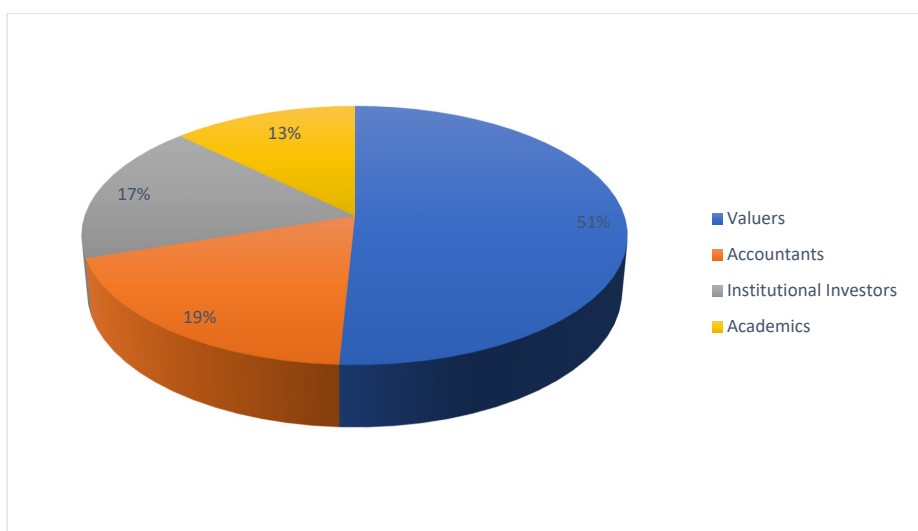

**Figure 1.** Demographic and Profile of the Respondents.

### 4.2. Analytical Hirerchical Process (AHP)

Table 3 shows the normalised weights of all factor groups. Ranking of the factors depends on the answers from the respondents of the questionnaire. Therefore, the weights of the factors calculated with the AHP method in this study are meant to demonstrate the purpose. It is important to state that making the pairwise comparisons by the professionals in the industry that filled out the questionnaire is going to provide expertise, especially in business valuation cases, as the comparisons will be project-specific. According to the AHP study results, the normalised weights of the six major factor groups change from 0.041 to 0.112. The lowest value 0.041 corresponds with the "market approach", and the maximum value 0.112 corresponds with the "professional liabilities". The second rank with the value 0.107 relates to the "process", followed by "asset-based approach" (0.084) and "discount rate" (0.055). In order to ensure that the results pass the consistency test, they must meet the standard of the AHP weighting test that has been assessed. The data were used to calculate the relative weighting values of the seven assessment dimensions. The reliability of the questionnaires was confirmed based on the consistency ratio standard which is >0.1. For the purpose of validity, the panel of focus group discussion (FGD) experts was consulted for seven different dimensions. From the respondents' responses, most sub-criteria were identified and were selected and weighted as the main criteria in AHP.

**Table 3.** Normalized Weight of Risk Groups.

| No. | Factor Groups | Normalised Weight |
|-----|---------------|-------------------|
| 1. | Professional liabilities | 0.112 |
| 2. | Process | 0.107 |
| 3. | Asset-based approach | 0.084 |
| 4. | Discount rate | 0.055 |
| 5. | Income approach | 0.042 |
| 6. | Fair value | 0.044 |
| 7. | Market approach | 0.041 |

### 4.3. Business Valuations Approached

Table 4 presents the decision hierarchy based on critical factors in business valuation approaches in determining the value. This test is aimed to assess the efficacy of the model to three case studies based on the questionnaires. These cases represent varying degrees of risk in business valuation intensity. This technique evaluated each case using the AHP procedure and the weights from the general model and was paired with the category of professionals comprised of valuers, accountants, and institutional investors. This is conducted for the purpose of assessing the risk in business valuations according to case study methods as well as groups of professionals. Academics were excluded from this group due to them having no practical experience engaged in real cases of business valuations. In order to apply the model into three cases, the comparison of each case, namely, asset-based, income, and market approach, was performed. The case comparison process is the same as the attribute criteria process which was discussed in an earlier section.

**Table 4.** Assessment of Case Studies' Values Based on Group Respondents.

| Factor Groups | Intensity of Importance in Business Valuation Risk | | |
|---------------|---------|-------------|------------------------|
| | Valuers | Accountants | Institutional Investors |
| Asset-based Approach | 5.34 | 6.29 | 5.34 |
| Income Approach | 7.31 | 7.45 | 8.32 |
| Market Approach | 8.79 | 9.43 | 7.34 |

The analysis revealed that for the asset-based approach, accountants perceived high risk compared to other professionals. Interestingly, both valuers and institutional investors



observed similar levels of risk. The findings signified that most accountants place risk in asset-based approaches, such as the existence of related party transactions, accounting disclosure, estimation of an asset's value, evaluation industry prospects, and a top-down forecasting approach. These are among the risks that need to be highlighted while determining the value in business valuation cases if the valuers are using this approach.

As for the income approach, all professional groups seem to place high concern in all areas of risk. Risk areas such as removing nonrecurring items, adjustment of depreciation, using the valuer's forecast, productive investment capacity to meet operational demands, the existence of non-operating assets, and the use of invested capital are among the research areas that professionals identify as areas that could expose valuers to risk in determining value in business valuation cases. Among these professionals, institutional investors have high concern about the risk in this approach, followed by accountants and valuers.

Market approach is the fundamental approach in determining fair value because this approach is derived from the current market situation. Therefore, the risk areas in this approach provided more concern among accountants, which highlights that this approach could expose valuers to risk areas. Risk areas such as legal risk, regulatory risk, business risk, market risk, economic risk, product risk, operating risk, asset risk, financial risk, and technological risk could be the areas which are exposed to the valuers. Likewise, all groups of professionals who were respondents were also concerned by this approach. From the FGDs, the major concern for this approach was due to the unpredictable market situation in recent years. Several events, such as financial crises, health crises, the political situation, an imbalance in housing demand and supply, the oil price crisis, and the Industrial Revolution 4.0, could be the factors that have a significant impact on the market situation. In conclusion, the assessment of case studies based on the three major respondents in this research revealed that all groups of professionals have major concerns in almost all risk areas of all types of valuation approaches in business valuations. The market approach appeared to be the highest risk area which is highlighted from the feedback of the respondents. An unprecedented market situation in recent years prompted the professionals to highlight these risk areas in the market approach that could be exposed to valuers.

The scores for each factor on other risk sources can be worked out and provide the level of risk based on each factor. The model also has its objectivity and stability and its applicable value to assess the risk level for business valuation for professional liability and risk mitigation. Hence, the scores for the matrix have a high risk for all factors across all experts. The findings also revealed that the process factor was verified to be the highest factor compared to method and profession. Therefore, in other words, it can be summarized that valuers need to highlight in order to determine the value in business valuation, that process methods could expose valuers to high risk compared to the other factors.

Results of the relative weights are presented in Table 5 under different criteria. For professional liabilities, the level of weight is between 13.67 to 39.53 which has an overall weightage from 4.29 to 19.53. The first rank from the overall sequence is "limited benchmarking data" followed by "include stipulation limiting" and "technology". From the respondents' feedback, it seems that most of them have the view that, under professional liabilities, factors related to the data will expose business valuer practitioners to risk. Previous research by Newell (2010) in assessing quality valuation reports also found that data represent one of the major factors in producing quality valuation reports. Furthermore, technology seems to be one of the factors that expose valuers to risk in terms of professional liabilities. Previous findings by Armugam et al. (2007) also revealed that most Malaysian valuers were still low in implementing technology to determine the valuation process. In similar research findings, Sunderajoo (2017) also found the low increase percentage from previous findings only in terms of implementing technology in the valuation process. Although these findings were based on the general valuation process, nevertheless, it reflects the business valuation as well due to the complexity and new establishment of this concept in Malaysia.

**Table 5.** Relative Weights of Major Criteria and Minor Criteria.

| Criteria | Sub-Criteria | Level | Overall | Overall Sequence |
|---|---|---|---|---|
| Professional Liabilities | Limited benchmarking data | 39.52 | 19.53 | 1 |
| | Include stipulation limiting | 32.42 | 17.53 | 2 |
| | Technology | 43.18 | 15.43 | 3 |
| | Conflict of interest | 32.51 | 13.53 | 4 |
| | Ethics related to questions | 25.31 | 9.34 | 5 |
| | Working with other professionals | 18.74 | 8.53 | 6 |
| | Statement agreement letters | 32.54 | 7.63 | 7 |
| | Validation of reports | 23.54 | 6.32 | 8 |
| | Check on the website against overstating | 15.39 | 5.25 | 9 |
| | Identify common trouble spots | 13.67 | 4.29 | 10 |
| Quality Valuation Report | Investment recommendation | 15.89 | 12.78 | 1 |
| | Information to allow a reader to critique the valuation | 17.45 | 11.98 | 2 |
| | Conflicts of interest | 11.45 | 10.42 | 3 |
| | Distinguishes clearly between facts and opinions | 24.43 | 10.32 | 4 |
| | Timely information | 21.87 | 9.43 | 5 |
| | Implications of the asset valuation | 15.98 | 9.21 | 6 |
| | Writing quality reports | 32.45 | 8.56 | 7 |
| | Analysis, forecasts, valuations, and recommendations that are all internally consistent | 18.67 | 8.41 | 8 |
| | Key assumptions clearly identified | 28.55 | 7.23 | 9 |
| Discount Rate | Evaluating adjusted discount rate | 11.45 | 6.82 | 1 |
| | The reliability of accounting fair value | 14.28 | 6.31 | 2 |
| | Direct relationship between asset prices and fair value to shareholders does not always occur | 11.34 | 5.32 | 3 |
| | Unrealised holding losses gains and | 8.13 | 5.19 | 4 |
| | Calculation of cash flow | 9.49 | 4.53 | 5 |
| | The relevance of accounting measures fair value | 12.87 | 4.19 | 6 |
| | Recognition of unrealised holding gains and losses | 9.43 | 4.12 | 7 |
| | The lack of information and data transparency | 6.45 | 4.12 | 8 |
| | Calculation in discount rate | 8.43 | 3.67 | 9 |
| | Determine required rate of return | 8.92 | 3.16 | 10 |
| | Determine discount rate | 7.12 | 2.42 | 11 |
| | Determine cost of capital | 6.54 | 2.13 | 12 |
| Fair Value | Direct relationship between asset prices and fair value to shareholders does not always occur | 22.53 | 14.43 | 1 |
| | Recognition of unrealised holding gains and losses | 21.45 | 11.45 | 2 |
| | The reliability of accounting fair value | 18.45 | 11.39 | 3 |
| | Unrealised holding losses gains and | 19.49 | 10.48 | 4 |
| | The relevance of accounting measures Fair value | 15.56 | 10.38 | 5 |
| | The lack of information and data transparency | 15.64 | 9.45 | 6 |
| Income Approach | Removing the nonrecurring items | 24.98 | 14.31 | 1 |
| | Adjustment of depreciation | 27.42 | 13.98 | 2 |
| | Using the valuer's forecast | 26.59 | 12.84 | 3 |
| | Productive investment capacity to meet operational demands | 21.40 | 12.42 | 4 |
| | The existence of non-operating assets | 21.55 | 11.58 | 5 |
| | The use of invested capital | 21.56 | 11.39 | 6 |
| | Adjusted related party transactions | 22.39 | 10.53 | 7 |
| | An excess of net working capital | 19.78 | 10.48 | 8 |
| | Removing the effect of the debt | 25.89 | 10.43 | 9 |
| | Evidence of underutilised capacity | 24.92 | 9.48 | 10 |
| | Adjustment of normalisation | 18.54 | 9.45 | 11 |
| | Eliminating nonrecurring income | 23.80 | 9.44 | 12 |
| | Insufficient management or employee skills or capacity | 19.45 | 9.40 | 13 |

**Table 5.** *Cont.*

| Criteria | Sub-Criteria | Level | Overall | Overall Sequence |
|---|---|---|---|---|
| Market Approach | Legal risk | 31.45 | 14.29 | 1 |
| | Regulatory risk | 27.34 | 13.51 | 2 |
| | Business risk | 22.56 | 13.04 | 3 |
| | Market risk | 22.78 | 12.65 | 4 |
| | Economic risk | 24.89 | 12.38 | 5 |
| | Product risk | 25.41 | 11.49 | 6 |
| | Operating risk | 24.87 | 11.34 | 7 |
| | Asset risk | 22.78 | 10.76 | 8 |
| | Financial risk | 21.15 | 9.67 | 9 |
| | Technological risk | 21.43 | 9.49 | 10 |
| Asset-based Approach | Existence of related party transactions | 17.40 | 13.18 | 1 |
| | Accounting disclosure | 21.48 | 12.98 | 2 |
| | The estimation of an asset's value based on variables perceived to be related to future investment returns, or based on comparisons with closely similar assets | 15.29 | 12.40 | 3 |
| | Understanding the business including evaluation industry prospects, competitive positions, and corporate strategies all of which contribute to making more accurate forecasts | 21.28 | 11.47 | 4 |
| | Intrinsic value incorporated with the going-concern assumption that a company will continue operating for the foreseeable future | 15.29 | 11.38 | 5 |
| | Property defining the standard value | 1445 | 10.45 | 6 |
| | High management or director turnover | 1589 | 10.45 | 7 |
| | A top-down forecasting approach moves from macro-economic forecast to industry forecasts to individual company and asset forecast | 20.48 | 9.67 | 8 |
| | Two or more estimates of value for machinery and equipment | 15.98 | 9.41 | 9 |
| | Management pressures to meet debt covenant or earning expectations | 19.45 | 9.39 | 10 |
| | Obligation to provide substantive and meaningful content | 15.29 | 9.37 | 11 |
| | Reports through regulatory filings disputes with and/or changes in auditors | 17.76 | 9.19 | 12 |
| | Do not let leave it up to other valuers to provide a valuation to avoid inconsistency | 13.53 | 8.49 | 13 |
| | Material non-audit services performed by audit firms | 15.39 | 8.45 | 14 |
| | Economic, industry or company-specific pressures on profitability such as loss of market share or declining margins | 18.74 | 8.41 | 15 |
| | The intrinsic value of an asset is its value given hypothetically with complete understanding of the asset's investment characteristics | 14.29 | 8.41 | 16 |
| | Present value models of common assets are the most important types of absolute valuation models | 18.21 | 8.34 | 17 |
| | Fair value is the price at which an asset or liability would change hands if neither buyer nor seller were under compulsion to buy or sell and both were informed about material underlying facts | 18.31 | 8.31 | 18 |
| | Valuers must hold themselves to both standards of competence and standards of conducts | 17.31 | 7.57 | 19 |
| | The assumption that the market price of security can diverge from its intrinsic value as suggested by the rationale of efficient market formulation of efficient market theory | 13.28 | 7.41 | 20 |
| | Using two or more concepts to value machinery | 9.34 | 7.19 | 21 |
| | Existence of excessive offers, employer or director loans | 8.82 | 5.38 | 22 |
| | High management turnover or director | 9.34 | 4.31 | 23 |
| | Excessive pressure on company personnel | 10.87 | 4.18 | 24 |
| | A history of securities' law violations, reporting violations or persistent late filings | 7.44 | 4.18 | 25 |

Answers from the respondents also revealed that factors such as "validation of report", "check on the website against overstating", and "identify common trouble spots" were the criteria that had less risk of exposure to the valuers. This corresponds with the findings from the FGDs who mentioned that currently these factors are not crucial for valuers in order to limit the exposure of risk.

Under the sub-criteria of the quality valuation report, the weightage of level ranges from 15.89 to 28.55 which represents the overall weightage from 7.23 to 12.78. The results also show that "quality valuation report" relative weight is lower than "professional liabilities". Under the sub-criteria of "quality valuation report", the respondents' view places three major higher risk exposures to valuers, which are "investment recommendation", "information to allow a reader to critique the valuation", and "conflict of interest". These results are also similar to previous research by Newell (2010) which stated that clients are quite concerned about the valuation, especially related to the investment method. This is due to the lack of information and transparency of data which could lead to a less accurate value to the subject of property.

The findings from the relative weight show that the range is within 6.54 to 11.45 which represents the overall range value from 2.13 to 6.82. It is noted that the range of the value is close, which describes the risk areas in discount rate as very significant. Risk areas such as "evaluating adjusted discount rate", "the reliability of fair value accounting", and, "direct relationship between asset prices and fair value to shareholders/stakeholders does not always occur" were perceived as the most high-risk areas to the valuers (from 5.32 to 6.82). Although risk areas such as "determine cost capital" and "determine discount rate" ranked at the low areas, nevertheless, due to the close range of weightage value, valuers should consider these areas as significant areas that could expose valuers to risk.

The next sub-criteria from the relative weighing analysis are to assess risk areas in fair value. There are six criteria under fair value which range from 15.64 to 22.53 and represent an overall value range from 9.45 to 14.43. Several risk areas in fair value indicate high risk areas, namely, "direct relationship between asset prices and fair value to shareholders/stakeholders does not always occur", "recognition of unrealised holding gains and losses", and "the reliability of fair value accounting" which could expose valuers to high risk in the business valuation process. The wide range of value is 11.39 to 14.43 which signifies the professionals' perception that these areas have possibilities to occur. This research attempts to assess risk areas based on the case study approach by using different valuation methods. The use of fair value in business valuation is quite significant as it reflects the market's assessment of the effects of current market conditions, especially when using financial instruments rather than relying on historical cost accounting. Fair value in business valuation reflects the economic substance and market conditions. The findings reveal that several risk areas such as "direct relationship between asset practices and fair value to shareholder/stakeholder does not always occur", "recognition of unrealised holding gains and losses", and "the reliability of fair value accounting" are the areas that professionals in business valuations need to highlight in determining the value under business valuation cases.

The next case study area which needs to be assessed in determining risk areas for business valuation is the income approach. A total of 13 sub-criteria were identified under the income approach to identify the level of risk. In an income approach, it is often that valuers are not able to find good information about market multiples or capitalisation rates to apply to the company's benefit stream. Valuers normally try to compare the risk associated with the benefit stream to alternative types of investment in the marketplace. Therefore, valuers need to go a long way before having knowledge about the rates of return available in the marketplace. This research attempts to identify the risk areas in an income approach which could expose valuers to risk in the business valuation process. The survey amongst respondents revealed "removing the nonrecurring items", "adjustment of depreciation", and "using the valuer's forecast" as the risk areas in the income approach which have potential to expose valuers to high risk. The close weightage range between

12.84 and 14.31 indicates that these risk areas are significant to each other. Although several risk areas, namely, "adjustment of normalisation", "eliminating of nonrecurring income", and "insufficient management or employee skills or capacities" recorded low weightage points, nevertheless, the closest point to the highest point showed that these areas also potentially expose valuers to risk in determining the value. Therefore, it could be summarised that professionals in business valuations perceived, in determining risk under the income approach, that all risk areas have the possibility of exposing valuers to risk. As a consequence, there is a need for valuers to highlight these factors while determining the value of the property subject matter.

## 5. Discussions

Overall, this research explored the risk mitigation from different perspectives, namely, professional liabilities, fair value, discount rate, quality valuation report, and case studies which were based on the valuation approaches. These are the general themes which aim to solve the problem statement which were identified earlier. Business valuation has become more complex with the competition of the business world together with challenges in economic obstacles which result in businesses needing to maneuver their direction to sustain and grow. Several business activities such as mergers, acquisitions, and hostile takeovers for monopoly domination of the market and business survival occur. As a result, the demand for business valuation services is emerging, and the professionals in these areas need to be equipped with knowledge not just specific to valuation and real estate but also other knowledge such as accounting and professional ethics. Certainly, these professionals need to provide an accurate business valuation service as well as knowledge and awareness of the risks that could occur during the process to determine business value. Risk mitigation in business valuation depends on the variety of business ownership, business legal status, the owner's level of asset responsibility, business existence, potential in change of ownership, type of management, goals of business valuation, etc. Although this research identified the category of research based on main activities in business valuation as well as valuation approaches, the risk could span across several factors. The general approach in risk mitigation has also been found in another research findings such as Sivitska and Makhmudov (2020) where the study identified that factors such business risk, operational risk, market risk, economic risk, industry risk, revenue growth, competition, diversification, employee relation are the main factors in determining risk in business valuation. Other studies include those by Ernst (2022), Scheurwater (2020), and Gleißner (2019).

In determining the value in business valuation cases, valuers need to consider the three most common valuation approaches, namely, the income-based approach, asset-based approach, and market- based approach. From the findings, it was revealed that, in the income approach, the most frequent method under this approach was the capitalisation of benefits method and discounted future benefits method which need to consider a variety of risk factors such as:

i.     Capitalisation of benefits
ii.    Normalisation adjustments
iii.   Nonrecurring revenue and expense items
iv.   Taxes
v.    Capital structure
vi.   Appropriate capital investment
vii.  Noncash items
viii. Qualitative judgements for risk used to compute discount and capitalisation rates
ix.   Expected in future changes
x.    Discounted future benefits
xi.   Forecast/ projection assumptions
xii.  Forecast/ projection earnings or cash flows
xiii. Terminal value

For an intangible asset, valuers should also consider the following areas:

i.      Remaining useful life
ii.     Current and anticipated future use of the intangible asset
iii.    Rights attributable future use of the intangible asset
iv.     Right attributable to the intangible asset
v.      Position of intangible asset in its life cycle
vi.     Appropriate discount rate for the intangible asset
vii.    Appropriate capital or contributory asset charge
viii.   Research and development or marketing expenses needed to support the intangible asset
ix.     Allocation of income to intangible asset
x.      Whether it involves tax amortisation
xi.     Discounted multi-years excess earnings

In the market approach, with consideration of the risk areas in a market approach, valuers must perform a comparative analysis of qualitative and quantitative similarities and differences between the guideline companies and the subject property to assess the investment attributes of the guideline companies relative to the subject property. Therefore, valuers should use only appropriate multiples based on the underlying financial data of each company's guidelines. Financial ratios for the guideline companies, as well as the comparative analyses of the qualitative and quantitative factors regarding the differences between the guideline companies and the appraisal subject, should be used to determine the appropriate valuation multiples to apply to the subject property. The risk factors that were identified seem to agree with the previous research conducted by Hanlin and Claywell (2010) which mentioned financial condition and the impact of external factors as the most important risks that need to be considered in the market approach in valuing business. Tajani et al. (2019) also emphasised that within the context of determining the market value of ordinary properties, where comparison data are generally available, the market technique can be used.

In the income approach, the findings and survey analyses revealed several risk areas that valuers need to highlight. Among the risk areas that will be exposed to valuers are removing the nonrecurring items, productive investment capacity to meet operational demands, and depreciation adjustment. These were also highlighted by Gallati (2022) which mentioned that one of the major problems in finance is the valuation and pricing of income streams. The problem appears simple on the surface, as it just entails determining the quantity and time of cash flows expected from retaining the claims and then discounting them back to the present. Furthermore, the income approach should also be reflected in the market value multiples in the market approach (Fishman et al. 2013).

The significant risk area in the asset-based approach is to relate the value to investment returns which could expose valuers to high risk due to the cost duplication business being appraised. This includes the evaluation of industry prospects that can reflect the economic balance sheet of the property subject in terms of high management, and it involves the data gathering from the high management level. The on-site interview is an essential part of the data-gathering phase of the appraisal engagement. Interviewing management at the company's facility has several advantages. First, seeing the facility's physical layout can ascertain such items as the capacity of the plant and working environment. Management will also feel more comfortable in its environment.

## 6. Conclusions

Overall, this research explored the risk factors in determining fair value in business valuation. Determining the value for the business in the companies requires numerous steps. Valuers will expose different types of risk, from general professional liabilities to details of risk, depending on the valuation approach. This research introduces extensive analyses using the Analytical Hierarchical Procedure (AHP) technique. Previous studies have not used this technique to explore the analyses from a quantitative point of view.

The findings from this research identified risk factors from the perspective of the business valuation approach. This research also analysed the business valuation within the scope of risk mitigation and fair market value from a local perspective.

Based on the AHP analysis, seven (7) factor groups were identified within the business valuation practice: professional liabilities, process, the asset-based approach, discount rate, the income approach, fair value, and the market approach. These areas are the critical risk factors for valuers in determining the value of business valuation. The analyses also depict the sub-criteria, explaining the significant essential risk factors. These include limited benchmarking data, stipulation limiting, investment recommendation, complete information, evaluating the adjusted discount rate, the direct relationship between asset prices and fair value, adjustment of depreciation, using the valuers' forecast, legal risk, regulatory risk, accounting disclosure, and the existence of related party transactions. Risk mitigation in business valuation depends on the variety of business ownership, business legal status, the owner's level of asset responsibility, business existence, potential in the change of ownership, type of management, goals of business valuation, etc. Although this research identified the category of research based on main activities in business valuation as well as valuation approaches, the risk could span across several factors.

Although this study is very extensive research in assessing the risk mitigation and fair market value in business valuations within the local context, further research will need to be conducted in order to provide a whole perspective of the Malaysian property valuers' profession. Further research will need to be conducted focusing on the accounting reports' risk and market value that reflect on business valuations. It would be interesting to assess accounting reports from the perspective of the literature and practical examples, as it will provide a whole perspective of risk in determining value for business valuation cases. Future research could also include global case studies for benchmarking processes. As risk and fair market value are a global valuation issue, benchmarking with global case studies and standards will therefore provide a clearer picture for Malaysian business valuation practices, especially in highlighting risk and fair market circumstances.

The demand for business valuation due to financial market uncertainties and economic and business challenges has driven professionals such as accountants, valuers, and lawyers into service business valuations. However, in Malaysia, every profession offers services for specific procedures according to their practice and suitability. As a result, offering a business valuation service is considered risky, and many of the professionals, exceptionally qualified and skilled valuers providing this service, refuse to continue with this practice. Therefore, initiatives are needed to establish professional liability risk when acquiring fair market value (FMV) in offering business valuation services. The demand for business valuation services is increasing, and these professional services are provided by experts from valuation or accounting backgrounds or a combination expertise of these professions. Indeed, these professionals must provide accurate business valuation services that benefit the client and fair value to third parties using their disclosure. Business valuations for listed companies are more transparent because the company's value can be determined with real financial market information, and the FMV method is commonly used. However, in the case of a takeover of unlisted or private companies, the data are challenging to obtain. Therefore, in determining FMV, the business valuation process should ensure that appropriate methods are implemented. Therefore, this research aims to highlight these issues using research tools to quantify all the risks involved in determining the business valuation. This research also attempts to take risks in business valuation more objectively and can be used as a guideline for practitioners in this field. The need to emphasise risk is significant due to challenging competition in the business world and the latest economic obstacles witnessed by the growing number of mergers, acquisitions, and hostile takeovers for monopoly market domination and business survival.

Consequently, the demand for business valuation services is increasing, and these professional services are offered by experts from valuation or accounting backgrounds or a combination of these professions. Indeed, these professionals must provide accurate

business valuation services that benefit the client and fair value to third parties using their disclosure. Business valuations for listed companies are more transparent because the company's value can be determined with real financial market information, and the FMV method is commonly used. However, in the case of a takeover of unlisted or private companies, the data are challenging to obtain. Therefore, in determining FMV, the business valuation process should ensure that appropriate methods are implemented. Moreover, the definition of FMV is very subjective. This research can address the two main issues in this study's subject matter: risk mitigation and FMV.

**Author Contributions:** Conceptualization, H.M.A. and R.A.J.; methodology, R.A.J.; software, M.N.R.; validation, M.N.R.; formal analysis, M.N.R.; investigation, M.N.R.; resources, K.A.; data curation, K.A.; writing—original draft preparation, M.N.R.; writing—review and editing, M.N.R. All authors have read and agreed to the published version of the manuscript.

**Funding:** This research received no external funding.

**Institutional Review Board Statement:** Not applicable.

**Informed Consent Statement:** Not applicable.

**Data Availability Statement:** Not applicable.

**Conflicts of Interest:** The authors declare no conflict of interest.

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
