# Peer review of "Identification of Risk Factors in Business Valuation"

_jrfm, doi:10.3390/jrfm15070282_

Round 1

Reviewer 1 Report

The manuscript present and interesting risk analysis case study. Managing business risks is a never-ending challenge. Method selection and the application level are appropriate, but some issues must be fixed before the publication:

  • The research and study goals should be highlighted.
  • The conclusions are too general. In line with the study goals, it must be rewritten.
  • Use more sub-chapters, especially in section 4 could be added for a better understanding. I could help to follow the message of the method better.
  • However, the application of the method seems to be correct, but some parts need more clarification, including the data source and survey.
  • Fewer details of the basics of AHP method could be enough in the literature review.

Author Response

The manuscript present and interesting risk analysis case study. Managing business risks is a never-ending challenge. Method selection and the application level are appropriate, but some issues must be fixed before the publication:

  • The research and study goals should be highlighted.
    • We add one more paragraph at section 1.0 to highlight the research and study goals
  • The conclusions are too general. In line with the study goals, it must be rewritten.
    • We rewritten the whole part of the conclusion section 
  • Use more sub-chapters, especially in section 4 could be added for a better understanding. I could help to follow the message of the method better.
    • We have revise section 4 to be divided into more sub-chapters
  • However, the application of the method seems to be correct, but some parts need more clarification, including the data source and survey.
    • We dedicate one sub- chapter to explaints the data source, data collection and survey
  • Fewer details of the basics of AHP method could be enough in the literature review.
    • We have simplified the details of AHP technique 

Reviewer 2 Report

Weaknesses of the paper: methodology, interpreting research results, discussion

The topic is interesting, but the paper has flaws and is accepted only after major revisions. I have some thoughts for possible improvement which the authors may consider:

My recommendation, it is necessary to modify the structure and content of the paper.

The paper should be prepared using the guidelines.

The paper should be divided into 1. Introduction, 2. Materials and Methods, 3. Results, 4. Discussion.

The Theoretical framework, … should be as a part of the introduction (not as a part of the methods). Introduction should end up by giving clues of the organization of the text.

The research methodology should be in chronological order. Please, transparently present methodology (research methodology) and transparently explain the results of the research.

And discussion? The discussion is not adequately described. Discussion should be the evaluation of the obtained results (author's original thoughts) in the light of the previous research (could include the items as explanation of the hypothesis).

Conclusions could be as guidelines to future research, especially the significance for the science.

+ see tables, formulas

Author Response

Weaknesses of the paper: methodology, interpreting research results, discussion

The topic is interesting, but the paper has flaws and is accepted only after major revisions. I have some thoughts for possible improvement which the authors may consider:

My recommendation, it is necessary to modify the structure and content of the paper.

  • The paper should be prepared using the guidelines.The paper should be divided into 1. Introduction, 2. Materials and Methods, 3. Results, 4. Discussion
    • We have revise the structure of the paper based on the reviewers’ reccomendation
  • The Theoretical framework, … should be as a part of the introduction (not as a part of the methods). Introduction should end up by giving clues of the organization of the text.
    • We have change the word ‘framework’ in the manuscript as this paper not aim to discuss the development of framework. Table 3 presents the factors group which extracted from the analysis by using AHP method
  • The research methodology should be in chronological order. Please, transparently present methodology (research methodology) and transparently explain the results of the research.
    • We have revised the methodology section to make more into cronological order. Consequently it also present the tranparent of methodology
    • We dedicate another section (discussion) to explain further the results of the research.
  • The discussion is not adequately described. Discussion should be the evaluation of the obtained results (author's original thoughts) in the light of the previous research (could include the items as explanation of the hypothesis).
    • We add one more section dedicate on the discussion part to give more evaluation aspect from the results.
  • Conclusions could be as guidelines to future research, especially the significance for the science.
    • We have rewritten the whole conclusion section to give highlight on future research 

Round 2

Reviewer 1 Report

I maintain the opinion about the benefits of the paper. The authors followed the instructions of the review. The improved version of the manuscript gives better emphasis to the results. The response mentions that Section 4 is divided into sub-chapters, but I can only see new paragraphs. Sub-sections must have a heading and a title. Please, add these!

Author Response

  1. Weaknesses of the paper: structure of the paper
  • I see improvements in the paper, but it is necessary to modify the structure of the paper.We have restructure the papers to give clear flow in explaining the subject matter

  1. The paper should be prepared using the guidelines.

          The paper should be divided into 1. Introduction, 2. Materials and Methods,    3. Results, 4. Discussion.

  • We have reorganise the structure of the paper according to the guidelines

Reviewer 2 Report

Weaknesses of the paper: structure of the paper

I see improvements in the paper, but it is necessary to modify the structure of the paper.

The paper should be prepared using the guidelines.

The paper should be divided into 1. Introduction, 2. Materials and Methods, 3. Results, 4. Discussion.

+ see tables, formulas

Author Response

  1. I maintain the opinion about the benefits of the paper.
  • We have explain the benefits of the paper at the end of the conslusion section

  1. The authors followed the instructions of the review. The improved version of the manuscript gives better emphasis to the results. The response mentions that Section 4 is divided into sub-chapters, but I can only see new paragraphs. Sub-sections must have a heading and a title. Please, add these!
  • We have reorganise section 4 into several sub- chapters.